# Feasibility Study of Single-Port Laparoscopic Techniques for Pancreatic Exploration, Ultrasound, and Biopsy in Dogs

**DOI:** 10.3390/ani15050652

**Published:** 2025-02-24

**Authors:** Changwoo Jeong, Kangwoo Yi, Sangjun Lee, Yong Yu, Suyoung Heo

**Affiliations:** College of Veterinary Medicine, Jeonbuk National University, Iksan 54596, Republic of Korea; vetnol@naver.com (C.J.); dlrkddn430@naver.com (K.Y.); sangjunlee@jbnu.ac.kr (S.L.); skyyong07@gmail.com (Y.Y.)

**Keywords:** single-port laparoscopy, laparoscopic ultrasound, pancreatic biopsy, minimally invasive surgery

## Abstract

This study evaluated a minimally invasive single-port laparoscopic technique to explore and sample the pancreas of dogs. The pancreas was examined laparoscopically through a small incision. Although most of the right lobe and body of the pancreas were accessible, the left lobe could not be reached because of its anatomical position. After exploration, a specialized laparoscopic ultrasound probe was used to evaluate accessible regions of the pancreas, successfully identifying key anatomical structures, such as blood vessels and ducts. Finally, a small tissue sample was collected from the right lobe for further analysis. This technique minimizes surgical trauma, reduces pain, and allows for faster recovery than conventional surgical methods. Although this study was performed in healthy dogs and biopsies were limited to specific areas, the findings demonstrate the potential for single-port laparoscopy to improve diagnostic procedures in veterinary medicine. This approach may benefit pets by providing safer and less-invasive options for the diagnosis of pancreatic conditions.

## 1. Introduction

Minimally invasive surgical techniques, in particular laparoscopic procedures, are increasingly being adopted in modern veterinary surgical practice due to their numerous clinical advantages [1,2,3,4]. Laparoscopy has been applied in various abdominal procedures, including organ exploration, tissue biopsy, and therapeutic interventions, as it reduces surgical trauma and promotes faster recovery [4]. Laparoscopic biopsy is widely utilized in small animals as a diagnostic tool, demonstrating both its feasibility and high diagnostic accuracy [5,6,7,8]. Previous studies have shown that laparoscopic biopsies yield sufficient tissue samples for histologic evaluation while lowering the risks of hemorrhage and tissue destruction, which are common complications of percutaneous needle biopsies [6,7]. Recently, laparoscopic pancreatic biopsy has emerged as a promising minimally invasive alternative, offering diagnostic sensitivity comparable to open surgery without increasing postoperative complications [7,8].

Despite these advances, laparoscopic pancreatic exploration and biopsy continue to present significant challenges. The high cost of specialized equipment, prolonged operative time, and steep learning curve required to master these techniques remain significant barriers [9]. Furthermore, these challenges are exacerbated by the deep anatomical location of the pancreas, which complicates access and visualization [10]. Successful outcomes also rely on careful patient selection, because anatomical and pathological variations can affect the feasibility and safety of laparoscopic procedures. Moreover, lesions identified on preoperative imaging may sometimes be undetectable during laparoscopy, complicating surgical planning and execution [11].

Laparoscopic ultrasonography (LUS) has become an indispensable tool in pancreatic surgeries, providing real-time imaging of deep-seated lesions and vascular structures that are not visible using conventional laparoscopic visualization techniques [12]. This modality bridges the gap between preoperative imaging and intraoperative findings, thereby enhancing surgical precision and safety [13].

The single-port laparoscopic technique (SPLT) is a recently developed approach in minimally invasive surgery that differs from conventional multi-port methods by requiring only a single incision and eliminating the need for multiple trocars. This technique provides several advantages, including improved cosmesis, reduced morbidity associated with peripheral ports, and potentially shorter hospital stays, while also simplifying surgical procedures and lowering the risk of infection and intraoperative hemorrhage by minimizing trocar usage [14,15].

Notably, SPLT allows for seamless conversion to other minimally invasive approaches, preserving the benefits associated with minimally invasive surgery. Recent applications of SPLT include cryptorchidectomy, cholecystectomy, adrenalectomy, splenectomy, hysterectomy, and hernia repair, with a growing interest in its efficacy compared to established minimally invasive methods [16,17,18].

The objective of this study was to evaluate the feasibility of single-port laparoscopic pancreatic exploration and biopsy in dogs and the application of LUS to the pancreas. This study aimed to determine whether this minimally invasive technique could effectively access and enable tissue sampling from lesions in the right lobe, left lobe, and body of the pancreas while minimizing complications and providing valuable procedural insight. Furthermore, the integration of LUS was analyzed to assess its ability to enhance real-time imaging of the pancreatic parenchyma and vascular structures.

## 2. Materials and Methods

### 2.1. Animals

This study involved six healthy Beagles and was approved by the Institutional Animal Care and Use Committee (IACUC) of Jeonbuk National University (JBNU IACUC, approval number: NON2024-085-002). The dogs were housed and maintained under the supervision of the Department of Veterinary Surgery at JBNU in a controlled environment with appropriate veterinary care, in compliance with national and institutional ethical guidelines. The study adhered to the principles outlined in the Guide for the Care and Use of Laboratory Animals and the ethical standards established by the JBNU IACUC. All dogs fasted for at least 8 h prior to surgery.

Following completion of the study, all animals were closely monitored during postoperative recovery under veterinary supervision. No dogs were euthanized and all procedures were performed in accordance with the humane endpoints established by the JBNU IACUC to minimize discomfort and ensure animal welfare.

### 2.2. Anesthesia

The dogs were premedicated with medetomidine (5 ug/kg, IV) and butorphanol (0.2 mg/kg, IV). For prophylaxis, cefazolin (22 mg/kg, IV) was administered 30 min before incision. General anesthesia was induced with propofol (4 mg/kg, IV) and maintained with sevoflurane.

### 2.3. Surgical Procedures

All laparoscopic procedures were performed by a single faculty surgeon, assisted by 1–2 experienced assistants and an anesthesiologist (Y.Y.). To allow for the potential conversion to open abdominal surgery, the area from the xiphoid to the pubis was shaved and surgically prepared. The animals were positioned on a tilting device with a 30-degree left lateral tilt (Figure 1).

#### 2.3.1. Pancreatic Exploration

A single port (Lapsingle; Sejong Medical, Paju, South Korea), with three 5 mm ports and one 10 mm port, was placed through a paramedian incision between the umbilicus and the right inguinal region to ensure adequate visualization and working space (Figure 2A–C). Initially, general exploration of the pancreas was planned to proceed in the following sequence: right lobe, body, and left lobe (Figure 2D). The exploration time was documented and defined as the duration from initiation of the procedure until the surgeon determined that all identifiable regions of the pancreas had been thoroughly examined and the exploration was complete.

#### 2.3.2. Laparoscopic Ultrasound

After exploration, an LUS transducer (LK51, Fujifilm, Tokyo, Japan) and articulating laparoscopic instrument (Artisential, LIVSMED, Seongnam, Korea) were introduced through the 10 mm port of the single port. An articulating laparoscopic instrument was used to hold the handle of the LUS transducer, enabling use of the probe at various angles. The pancreatic parenchyma was evaluated using ultrasound, along with identifiable areas during abdominal exploration (Figure 2E). To avoid iatrogenic damage of pancreatic parenchyma, the scan was conducted as gently as possible. As the transducer reached the pancreatic body, additional scanning of the duodenum was performed to visualize the major duodenal papilla (MDP). In addition, the color Doppler mode was used to identify the cranial pancreaticoduodenal artery and vein. A strict 3-minute time limit was set for each anatomical landmark; if this time was exceeded, the attempt was deemed a failure and the scan proceeded to the next structure. Additionally, LUS time was recorded, defined as the duration from the beginning of the ultrasound evaluation to the completion of the examination of all identifiable pancreatic regions, including the MDP, cranial pancreaticoduodenal artery, and vein (Figure 2F).

#### 2.3.3. Laparoscopic Pancreatic Biopsy

Babcock forceps and a bipolar vessel-sealing device (VSD; MarSeal 5 Plus, KLS Martin Group, Tuttlingen, Germany) were inserted into the abdominal cavity through a single port for pancreatic biopsy. The tip of the right lobe of the pancreas was gently grasped using Babcock forceps and the vascular structures were assessed. A biopsy was then performed by securing a 10 × 10 mm section with the VSD. The biopsy site was inspected for additional bleeding (Figure 2G–I). 

### 2.4. Closure

After completing all procedures, the port was removed and the gas in the abdominal cavity was evacuated. The external, internal oblique, and transverse abdominis muscles were sutured together in a single line using a simple continuous pattern (PDS II 2-0; Ethicon, Cornelia, GA, USA). The subcutaneous tissue was closed using a simple continuous pattern (PDS II 3-0; Ethicon, Cornelia, GA, USA) and the skin was sutured using a simple interrupted pattern (Nylon 3-0; Ailee, Busan, Republic of Korea). The incision length was measured after suturing (Figure 2J).

### 2.5. Postoperative Care

The dogs were postoperatively monitored for at least 3 h to ensure their safe recovery. Butorphanol (0.2 mg/kg IV) and meloxicam (0.2 mg/kg, SC) were administered immediately after surgery, followed by oral administration of 0.1 mg/kg once daily for analgesia. A dressing was applied to the incision site daily.

### 2.6. Data Collection

All data were analyzed using GraphPad Prism software (version 9.0.0; GraphPad Software, San Diego, CA, USA). Given the small sample size and non-normal distribution of the data, median data and range were used to summarize the study.

## 3. Results

Six spayed female Beagles participated in this study. The median age was 13 months (range: 12–30 months) and the median weight was 8.81 kg (range: 7.84–10.64 kg). The dogs were healthy and had no concurrent medical conditions.

### 3.1. Pancreatic Exploration

The median time from incision to portal placement was 319 s (range: 276–368 s). The median time for right lobe examination was 239 s (range: 204–268 s). The median time for examination of the pancreatic body was 370 s (range: 328–457 s). As per the surgeon’s judgment, complete exploration of the left lobe was deemed too challenging and exploration was discontinued in all cases (Table 1).

### 3.2. Laparoscopic Ultrasound 

The median LUS duration was 838 s (range: 729–878 s). With the exception of one case, ultrasonographic evaluation successfully identified the major anatomical landmarks, including the MDP, cranial pancreaticoduodenal vein, and cranial pancreaticoduodenal artery. In the case of Dog 1, exploration was terminated based on the surgeon’s subjective judgment that identifying the MDP was too challenging; therefore, only the cranial pancreaticoduodenal artery and vein were documented (Table 2 and Figure 3).

### 3.3. Pancreatic Biopsy

Tissue sections approximately 10 × 10 mm in size were successfully obtained from each dog. The biopsied tissue was retrieved through a single port and inspected to confirm that it was representative of the pancreatic parenchyma. Hemostasis was verified after each biopsy. In all six dogs, no additional bleeding was observed at the biopsy site upon withdrawal of the instruments.

### 3.4. Incision Length

The initial planned incision was 2.5 cm and the median length after skin suture was 2.8 cm (range: 2.6–3.0 cm).

### 3.5. Postoperative Outcomes

All six dogs exhibited signs of normal recovery from anesthesia. No cases of dehiscence, swelling, or erythema were observed at the surgical sites. The stitches were removed 10 days after the surgery.

## 4. Discussion

The single-port laparoscopic approach demonstrated feasibility and effectiveness for the exploration and biopsy of the right lobe and body of the pancreas. Portal placement and exploration time for the right lobe and body were consistent in all cases, underscoring the reliability of this minimally invasive technique. Furthermore, the integration of LUS enabled real-time visualization of the pancreatic parenchyma and major vascular structures, suggesting its potential to enhance intraoperative precision and facilitate more accurate localization of lesions in future applications. The port placement strategy employed in this study deviated from conventional techniques and was specifically designed to accommodate the characteristics and morphology of the pancreas. The port was strategically placed based on the anatomical morphology of the pancreas to facilitate optimal ultrasound application across all regions. The selected site facilitated effective visualization of the right lobe and body of the pancreas, ensuring consistent and reliable access during exploration. Additionally, this placement allowed for the convenient application of a laparoscopic ultrasound on the right lobe and body. However, complete visualization of the left pancreatic lobe was not achieved in any of the dogs, likely because of anatomical constraints imposed by the omentum and the deep location of the pancreas. Minor bleeding associated with muscle incision during port placement was observed in some cases, but was effectively managed with electrocautery or resolved spontaneously without adverse outcomes. Although this study employed a single-port approach, multi-port laparoscopic techniques, including three- or four-port configurations, have been successfully utilized in prior studies of distal pancreatectomy and mass removal in cats and dogs [19,20]. These multi-port methods have proven advantageous for optimized port placement and use of additional retraction tools that facilitate improved exposure of the pancreas, including the left lobe. These findings indicate that modifications in port placement or the integration of additional instruments may enhance the practicality and efficiency of the single-port approach. Thus, employing hybrid techniques, such as introducing a retraction port in the left subcostal region, may further optimize visualization and facilitate improved access to the left lobe of the pancreas [19]. Although the current results have some limitations, combining single-port methods with flexible instruments can enhance access to difficult regions. Additionally, strategically adding a secondary port could improve visualization and instrument handling and optimize surgical outcomes [21]. The disadvantage of using a single port is that the instruments may collide with each other, which is common in restricted working spaces, in particular when accessing deep-seated organs such as the pancreas. However, a single port consists of a flexible material that allows for a substantial degree of flexibility in the manipulation of laparoscopic instruments. Furthermore, a resection may not be difficult for surgeons with experience performing laparoscopies. In this study, the use of articulating instruments significantly enhanced maneuverability and precision within a limited working space. These instruments facilitated smoother manipulation, effectively mitigating technical challenges commonly encountered in single-port laparoscopic procedures [22]. Their application not only improved ease of surgical handling but also likely contributed to the efficiency and accuracy of the resections, especially in such an anatomically complex region [23]. LUS evaluation played a key role in this study, enabling the accurate identification of the pancreatic parenchyma and essential anatomical structures, including the MDP, cranial pancreaticoduodenal artery, and cranial pancreaticoduodenal vein. Of note, the cranial pancreaticoduodenal vein, a landmark commonly utilized in routine abdominal ultrasonography, served as a reliable reference point during LUS-guided assessments [24,25]. These findings highlight the utility of LUS in evaluating the resection margins of pancreatic masses and in detecting lesions that, while apparent on preoperative imaging, may not be visible during direct laparoscopic inspection. By effectively integrating preoperative diagnostic insight with real-time intraoperative imaging, LUS has the potential to improve the precision of lesion localization, support intraoperative decision-making, and contribute to the success of minimally invasive pancreatic surgeries. Articulating laparoscopic instruments allow for optimal probe positioning, thereby enhancing the application of ultrasound technology in challenging anatomical regions [26]. However, their use adds complexity to the procedure, requiring advanced surgical skills and underscoring the need for specialized training. Pancreatic biopsy was performed safely and effectively using this technique. All dogs underwent a biopsy of the distal portion of the right pancreatic lobe without complications and each tissue sample measured 10 × 10 mm in size. No additional bleeding or tissue damage was observed. However, as the study involved clinically healthy dogs, the distal portion of the right pancreatic lobe was specifically targeted for biopsy to minimize the risks. Additionally, histopathological analysis was not conducted on the biopsied tissues in this study, leaving the extent of marginal damage unassessed. The procedure was successfully performed without any perioperative or postoperative complications. All the dogs recovered safely and showed no signs of infection, dehiscence, or other adverse effects. The port-site incisions healed without issue and the sutures were removed on postoperative day 10, reflecting the minimal invasiveness and favorable healing observed with single-port techniques. This study has several limitations. First, as the experiment was conducted on clinically healthy dogs, its application to pathological conditions remains uncertain. The findings may not fully reflect the challenges encountered in diseased patients, necessitating further investigation in pathological cases. Second, the relatively small sample size of six dogs may limit the generalizability of the results. A larger-scale study is warranted to validate these findings and further assess the efficacy and feasibility of this approach in a broader clinical context. Third, a histological analysis of the biopsied pancreatic tissues was not performed, preventing the confirmation of pathological characteristics and limiting the ability to assess the potential diagnostic yield. Lastly, visualization of the left pancreatic lobe was not achieved in all cases, likely due to its deep anatomical location. Future studies should explore strategies to enhance access to this region, such as modifications in port placement or the integration of hybrid techniques.

## 5. Conclusions

This study demonstrates the feasibility and potential of single-port laparoscopic techniques for pancreatic exploration and biopsy in dogs. The integration of LUS enables real-time visualization and provides valuable insight into the pancreatic parenchyma and vascular structures. Future studies should validate this technique in pathological cases and explore strategies to overcome anatomical challenges, in particular, access to the left pancreatic lobe. Refining these aspects may facilitate its application in clinical cases involving pancreatic diseases, such as pancreatitis or tumors, as a less-invasive alternative to conventional methods.

## Figures and Tables

**Figure 1 animals-15-00652-f001:**
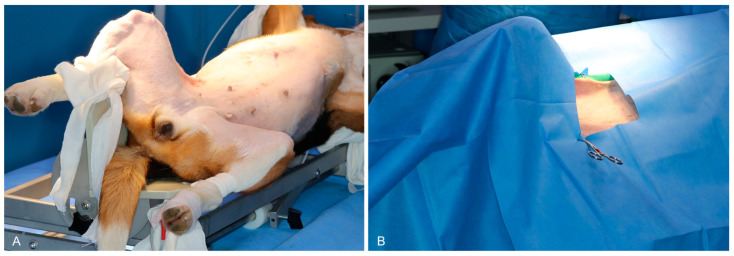
Positioning and surgical site preparation for single-port laparoscopic pancreatic exploration and biopsy in dogs. (**A**) Preoperative dorsal recumbent positioning of the dog in with its limbs secured to the surgical table, using a tilting device adjusted to 30 degrees to optimize visualization and access to the abdominal cavity. (**B**) Surgical draping to maintain sterility with the incision site prepped and marked for single-port laparoscopic entry under a tilted position.

**Figure 2 animals-15-00652-f002:**
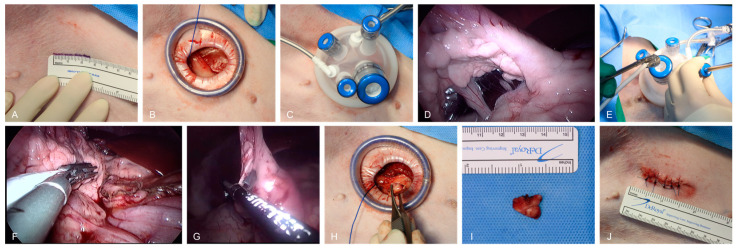
Stepwise procedure for single-port laparoscopic pancreatic exploration and biopsy in dogs. (**A**) The marking and preparation of the surgical site. (**B**) A small incision for single-port placement. (**C**) The placement of a single-port device with three 5 mm ports and one 10 mm port. (**D**) The insertion of articulating laparoscopic instruments and LUS transducer through the single-port system for real-time imaging. (**E**) The initial exploration of the abdominal cavity to locate the pancreas using laparoscopic guidance. (**F**) The handling of pancreatic tissue with articulating instruments during laparoscopic ultrasonography. (**G**) A biopsy of pancreatic tissue from the right lobe using a VSD. (**H**) The retrieval of a biopsied pancreatic tissue sample through the single-port system. (**I**) The pancreatic tissue sample. (**J**) The closure of the surgical site after evacuation of the abdominal cavity gas, showing incision length.

**Figure 3 animals-15-00652-f003:**
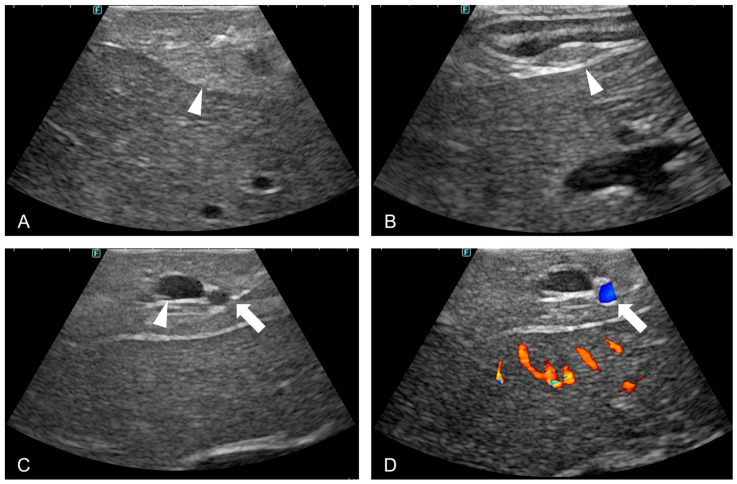
Ultrasound imaging of the pancreas and associated structures in dogs. (**A**) An ultrasound image displaying the echotexture of the pancreatic parenchyma (arrowhead). (**B**) An ultrasound visualization of the major duodenal papilla (arrowhead). (**C**) An ultrasound image identifying the cranial pancreaticoduodenal artery and vein (arrow). (**D**) A Doppler-mode ultrasound image confirming arterial pulsation and vascular flow in the cranial pancreaticoduodenal artery (arrow).

**Table 1 animals-15-00652-t001:** Surgical details of laparoscopic pancreas explorations and biopsies.

Dog	Body Weight, kg	Exploration Time, Right Lobe, s	Exploration Time, Right Body, s	Incision Length, cm
1	7.84	204	328	2.7
2	9.27	257	356	2.8
3	8.35	233	417	2.8
4	9.51	268	347	3.0
5	10.64	245	384	2.8
6	7.82	228	457	2.6

**Table 2 animals-15-00652-t002:** Details of laparoscopic pancreatic ultrasounds in this study.

Dog	Major Duodenal Papilla	Cranial Pancreatoduodenal Vein	Cranial Pancreatoduodenal Artery	LUS Time, s
1	✕	O	O	878
2	O	O	O	729
3	O	O	O	851
4	O	O	O	825
5	O	O	O	741
6	O	O	O	855

## Data Availability

The raw data supporting the conclusions of this study will be made available by the authors upon request.

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
