# Peer review of "Feasibility Study of Single-Port Laparoscopic Techniques for Pancreatic Exploration, Ultrasound, and Biopsy in Dogs"

_animals, 2025, doi:10.3390/ani15050652_

Round 1

Reviewer 1 Report

Comments and Suggestions for Authors

The authors focused their work on providing an interesting and promising approach combining single port laparotomy with ultrasound-guided organ identification for pancreas laparoscopic surgery. The experimental approach was well-designed and executed with outstanding results. The fact that it was developed in clinically healthy dogs does not affect the merit of the proposed innovation. This work will impact the surgical approach to laparoscopic surgery and overall minimally invasive surgery by the accuracy of ultrasound imaging, which will add to this successful surgical approach.

Author Response

Author’s reply to the review report

Reviewer 1

We sincerely appreciate the reviewer’s generous evaluation and insightful feedback. Your constructive comments have been invaluable in refining our manuscript, and we are grateful for your recognition of our work. Thank you for your time and thoughtful review.

Reviewer 2 Report

Comments and Suggestions for Authors

Dear Authors,

Thank you for choosing our journal for your article submission. I had the chance to review your manuscript, and I truly appreciate the effort and dedication you've put into your research on single-port laparoscopy and laparoscopic ultrasound in pancreatic surgery. Your work offers valuable insights that could positively influence advancements in the field.

While your study holds great promise, I suggest considering a major revision to address some key areas. These improvements will help clarify your findings and enhance their impact, increasing their suitability for publication. I encourage you to thoughtfully incorporate these suggestions during your revision process.

Please find my detailed review below.

Best Regards

Title and Abstract

- Strengths:
The title accurately reflects the study's focus on feasibility and minimally invasive techniques. The abstract is concise and effectively summarizes the methodology and key findings.

- Recommendations:
- Consider including a brief mention of the study’s limitations in the abstract for a more balanced perspective.
- Define clinically healthy dogs more precisely, specifying criteria such as age and weight.

Introduction

- Strengths:
The introduction gives a comprehensive background on laparoscopic techniques, their advantages, and the specific challenges linked to pancreatic surgery.

- Recommendations:
- Clarify the novelty of employing a single-port technique as opposed to existing multi-port methods.
- Include references to prior studies on laparoscopic biopsy in small animals, beyond general notes on minimally invasive surgery benefits.

Materials and Methods

- Strengths:
The methodology is detailed, facilitating the replication of procedures. The use of LUS for pancreatic assessment is well-supported.

- Recommendations:
- Justify the small sample size and discuss its potential impact on the generalizability of the findings.
- Provide a rationale for the choice of anesthesia and postoperative analgesics, specifically related to pancreatic exploration.
- Describe the training level or experience of the surgeon to assess procedural consistency.

Results

- Strengths:
Data are well-organized, with clear tables summarizing surgical times and outcomes.

- Recommendations:
- Consider including confidence intervals or standard deviations to better illustrate data variability.
- Address whether the inability to access the left lobe of the pancreas could be mitigated with alternative positioning or instruments.

Discussion

- Strengths:
The discussion effectively highlights the advantages of single-port laparoscopy and the integration of LUS.

- Recommendations:
- Discuss the biological significance of the findings, particularly in relation to ease of handling and visualization.
- Analyze potential complications in pathological cases more deeply.
- Expand on future research directions, offering specific suggestions for overcoming anatomical constraints.

Statistical Analysis

- Comment:
The method used is suitable for non-parametric data. However, considering more sophisticated methods or alternative analysis techniques could enhance the robustness of the findings.

Limitations

- Comment:
The small sample size and the absence of histological analysis are significant limitations that should be explicitly addressed in both the discussion and conclusion.

Conclusion

- Strengths:
The conclusion is concise and aligns well with the study's aims.

- Recommendations:
- Include a statement on the limited applicability to clinical settings without further validation.

Overall Recommendation: Major Revision
The manuscript presents valuable findings but requires substantial clarification of methodology, justification of choices, and a stronger discussion of limitations. These revisions will enhance its impact and readability for the audience.

Author Response

Author’s reply to the review report

Reviewer 2

<Title and abstract>

As per the reviewer’s suggestion, a statement regarding the study’s limitations has been incorporated to provide a more balanced perspective. Additionally, the criteria for defining ‘clinically healthy dogs’ have been further elaborated for clarity.

<Introduction>

As per the reviewer’s suggestion, we have clearly articulated the advantages of the single-port technique in comparison to the conventional multi-port approach. Additionally, we have restructured the content to align more precisely with the scope of the manuscript. Beyond general discussions on the benefits of minimally invasive surgery, we have incorporated and cited relevant prior studies specifically addressing laparoscopic biopsy in small animals.

<Material and Methods>

As per the reviewer’s suggestion, we have included additional details describing the surgeon’s training level and the measures taken to ensure procedural consistency. Furthermore, we have acknowledged the small sample size as a primary limitation of the live experiment and have integrated a discussion on its implications, including the need for further studies, within the discussion section. Regarding the surgical procedure, exploration, laparoscopic ultrasound, and biopsy were performed in all subjects, and accordingly, postoperative care was administered as described in the main text.

<Results>

As per the reviewer’s suggestion, we have expanded the discussion to include additional considerations regarding the observation of the left lobe of the pancreas. When using a single-port approach alone, visualization of the left lobe was challenging due to its coverage by other abdominal organs and the greater omentum. However, as suggested in other studies on laparoscopic pancreatic access, the use of a hybrid technique, such as the placement of an additional portal in the left cranial paramedian region, could facilitate improved visualization. Nevertheless, as this approach was not implemented in our study, we have discussed it as a consideration in the discussion section.

Additionally, as the data in this study did not follow a normal distribution, we presented the results using range instead of standard deviation.

<Discussion>

In accordance with the reviewers' suggestions, we have emphasized the aspects of visualization and ease of manipulation. Furthermore, building upon previous studies, we have expanded the scope of our research by proposing potential solutions for addressing the left lobe using hybrid techniques, such as the incorporation of an additional retraction portal. Regarding pathological cases, we have acknowledged this as one of the major limitations of our study, strongly indicating the necessity for further research.

<Limitation>

The limitations section explicitly addresses the small sample size and the absence of histological analysis, outlining their impact on the clinical significance of this study.

<Conclusion>

We have incorporated the points raised by the reviewer into the manuscript.

Reviewer 3 Report

Comments and Suggestions for Authors

The study's objective does not seem very clear to me. The laparoscopic approach to the pancreas in dogs has been widely studied and reported in cases of neoplasia and dysfunction. The authors argue that using single-port techniques could improve surgical results. However, they do not analyze any variable that seeks to clarify this idea. This is added to the fact that the tests were carried out on healthy animals to test an approach that does not seem novel in the specialists' community. Essentially, no benefit is observed from using a single port for this approach, and its feasibility is not novel.

The use of laparoscopic ultrasonography to describe structures and help in the intraoperative identification of lesions seems much more interesting. However, the work does not demonstrate its specific utility.

Despite being approved by a bioethics committee, I believe that the results of the research do not justify the use of healthy animals, especially the extraction of a biopsy that does not add anything to the global knowledge of the specialty and is associated with potential morbidities.

My suggestion is to rethink all the work surrounding the description of the use of laparoscopic ultrasonography in minimally invasive pancreatic surgery.

Author Response

Author’s reply to the review report

Reviewer 3

The objective of this study was to evaluate the feasibility of pancreatic exploration, biopsy, and the application of laparoscopic ultrasonography (LUS) using a single-port approach. To the best of our knowledge, there have been no prior studies utilizing laparoscopic ultrasonography for therapeutic purposes. The placement of the single port was determined based on the morphology of the pancreas and to facilitate the manipulation of the laparoscopic ultrasound probe.

Our findings indicate that access to the left lobe was challenging due to the presence of the omentum and the deep anatomical structures. However, based on previous studies, this limitation could potentially be addressed by incorporating an additional retraction portal caudal to the left rib, a modification that was not implemented in this study but has been discussed in the manuscript.

Furthermore, the application of laparoscopic ultrasonography in this study allowed for the identification of pancreatic parenchyma and anatomical landmarks. While we acknowledge the reviewer's concern that the absence of pathological clinical cases limits the assessment of its true utility, our findings demonstrate that LUS manipulation in the peripancreatic region was feasible and that anatomical structures could be clearly identified. These results highlight the need for further investigations to explore its clinical applicability.

In response to the reviewer's suggestion, we have strengthened the discussion on laparoscopic ultrasonography and made extensive revisions to the introduction and discussion sections accordingly.

Round 2

Reviewer 2 Report

Comments and Suggestions for Authors

Dear Authors,

Thank you for submitting the revised version of your manuscript titled "Feasibility Study of Single-Port Laparoscopic Techniques for Pancreatic Exploration, Ultrasound, and Biopsy in Dogs."

I appreciate the improvements made, particularly the clarifications regarding the methodology and the enhanced discussion of the study’s limitations. The manuscript is now clearer and more comprehensive.

I believe the revised version is suitable for publication in its current form.

Best regards,